# Ongoing Efforts to Improve Antimicrobial Utilization in Hospitals among African Countries and Implications for the Future

**DOI:** 10.3390/antibiotics11121824

**Published:** 2022-12-15

**Authors:** Zikria Saleem, Brian Godman, Aislinn Cook, Muhammad Arslan Khan, Stephen M. Campbell, Ronald Andrew Seaton, Linda Siachalinga, Abdul Haseeb, Afreenish Amir, Amanj Kurdi, Julius C. Mwita, Israel Abebrese Sefah, Sylvia A. Opanga, Joseph O. Fadare, Olayinka O. Ogunleye, Johanna C. Meyer, Amos Massele, Dan Kibuule, Aubrey C. Kalungia, Moyad Shahwan, Hellen Nabayiga, Giuseppe Pichierri, Catrin E. Moore

**Affiliations:** 1Department of Pharmacy Practice, Faculty of Pharmacy, Bahauddin Zakariya University, Multan 60800, Pakistan; 2Department of Pharmacoepidemiology, Strathclyde Institute of Pharmacy and Biomedical Sciences, University of Strathclyde, Glasgow G4 0RE, UK; 3Centre of Medical and Bio-Allied Health Sciences Research, Ajman University, Ajman 346, United Arab Emirates; 4Department of Public Health Pharmacy and Management, School of Pharmacy, Sefako Makgatho Health Sciences University, Molotlegi Street, Garankuwa, Pretoria 0208, South Africa; 5Centre for Neonatal and Paediatric Infection, St. George’s University of London, London SW17 0RE, UK; 6Health Economics Research Centre, Nuffield Department of Population Health, University of Oxford, Oxford OX1 2JD, UK; 7The Indus Hospital, Bedian Road, Lahore 54000, Pakistan; 8Centre for Epidemiology and Public Health, School of Health Sciences, University of Manchester, Manchester M13 9PL, UK; 9NIHR Greater Manchester Patient Safety Translational Research Centre, School of Health Sciences, University of Manchester, Manchester M13 9PL, UK; 10Queen Elizabeth University Hospital, Govan Road, Glasgow G51 4TF, UK; 11Scottish Antimicrobial Prescribing Group, Healthcare Improvement Scotland, Delta House, 50 West Nile Street, Glasgow G1 2NP, UK; 12College of Pharmacy, Yeungnam University, Daehak-Ro, Gyeongsan, Gyeongbuk 38541, Republic of Korea; 13Department of Clinical Pharmacy, College of Pharmacy, Umm Al-Qura University, Makkah 24382, Saudi Arabia; 14Department of Microbiology, Armed Forces Institute of Pathology, National University of Medical Sciences, Rawalpindi 46000, Pakistan; 15Department of Pharmacology, College of Pharmacy, Hawler Medical University, Erbil 44001, Iraq; 16Center of Research and Strategic Studies, Lebanese French University, Erbil 44001, Iraq; 17Department of Internal Medicine, Faculty of Medicine, University of Botswana, Private Bag 0713 UB, Gaborone 00704, Botswana; 18Pharmacy Practice Department, School of Pharmacy, University of Health and Allied Sciences, Volta Region, Hohoe PMB 31, Ghana; 19Department of Pharmaceutics and Pharmacy Practice, School of Pharmacy, University of Nairobi, Nairobi P.O. Box 19676-00202, Kenya; 20Department of Pharmacology and Therapeutics, Ekiti State University, Ado Ekiti 362103, Nigeria; 21Department of Medicine, Ekiti State University Teaching Hospital, Ado Ekiti 360211, Nigeria; 22Department of Pharmacology, Therapeutics and Toxicology, Lagos State University College of Medicine, Ikeja, Lagos 100271, Nigeria; 23Department of Medicine, Lagos State University Teaching Hospital, Ikeja 100271, Nigeria; 24South African Vaccination and Immunisation Centre, Sefako Makgatho Health Sciences University, Molotlegi Street, Garankuwa, Pretoria 0208, South Africa; 25Department of Clinical Pharmacology and Therapeutics, Hurbert Kairuki Memorial University, 70 Chwaku Road Mikocheni, Dar Es Salaam P.O. Box 65300, Tanzania; 26Department of Pharmacology & Therapeutics, Busitema University, Mbale P.O. Box 236, Uganda; 27Department of Pharmacy, School of Health Sciences, University of Zambia, Lusaka P.O. Box 50110, Zambia; 28Department of Clinical Sciences, College of Pharmacy and Health Sciences, Ajman University, Ajman 346, United Arab Emirates; 29Management Science Department, Strathclyde Business School, University of Strathclyde, 199 Cathedral Street, Glasgow G4 0QU, UK; 30Microbiology Department, Torbay and South Devon Foundation Trust, Lowes Bridge Torbay Hospital, Torquay TQ2 7AA, UK

**Keywords:** Africa, antimicrobials, antimicrobial stewardship programs, antimicrobial resistance, national action plans, quality indicators, strategies, surgical site infections, utilization patterns

## Abstract

There are serious concerns with rising antimicrobial resistance (AMR) across countries increasing morbidity, mortality and costs. These concerns have resulted in a plethora of initiatives globally and nationally including national action plans (NAPs) to reduce AMR. Africa is no exception, especially with the highest rates of AMR globally. Key activities in NAPs include gaining a greater understanding of current antimicrobial utilization patterns through point prevalence surveys (PPS) and subsequently instigating antimicrobial stewardship programs (ASPs). Consequently, there is a need to comprehensively document current utilization patterns among hospitals across Africa coupled with ASP studies. In total, 33 PPS studies ranging from single up to 18 hospitals were documented from a narrative review with typically over 50% of in-patients prescribed antimicrobials, up to 97.6% in Nigeria. The penicillins, ceftriaxone and metronidazole, were the most prescribed antibiotics. Appreciable extended prescribing of antibiotics up to 6 days or more post-operatively was seen across Africa to prevent surgical site infections. At least 19 ASPs have been instigated across Africa in recent years to improve future prescribing utilizing a range of prescribing indicators. The various findings resulted in a range of suggested activities that key stakeholders, including governments and healthcare professionals, should undertake in the short, medium and long term to improve future antimicrobial prescribing and reduce AMR across Africa.

## 1. Introduction

There are serious concerns globally with growing antimicrobial resistance (AMR), with an associated increase in morbidity, mortality and costs [1,2,3,4]. A recent study estimated that in 2019 alone there were 1.27 million deaths globally attributable to bacterial AMR and 4.95 million deaths associated with bacterial AMR, with the greatest burden in Western sub-Saharan Africa [2]. The high rates of AMR among African countries may reflect the fact that the greatest burden of all infectious diseases worldwide, including human immunodeficiency virus (HIV) and acquired immunodeficiency syndrome (AIDS), acute respiratory diseases, malaria and tuberculosis (TB), is currently in Africa [5,6,7,8,9], with associated prescribing of antimicrobials. This includes prophylaxis against opportunistic infections for patients with HIV/AIDS in view of their impact on morbidity and mortality [10]. Alongside this, high rates of inappropriate prescribing and dispensing of antibiotics across all sectors in Africa, including for viral infections such as acute respiratory tract infections, are exacerbated by appreciable purchasing of antibiotics without a prescription [9,11,12,13,14,15,16,17,18,19,20]. Self-purchasing of antibiotics is common across sub-Saharan Africa, enhanced by high patient co-payments, a lack of access to certain antibiotics coupled with the convenience of community drug stores and pharmacies as well as concerns with the public healthcare system alongside porous supply chains for medicines [11,12,21,22,23,24,25,26,27]. Having said this, Klein et al. (2019) in their recent study showed that whilst low- and middle-income countries (LMICs) had the highest rates of AMR, there was a less clear-cut link between consumption and resistance rates compared with high-income countries [28]. However, this study only involved South Africa among the African countries in its analysis, with currently very limited self-purchasing of antibiotics in South Africa with strict regulations [28,29]. There is also appreciable consumption of antibiotics in agricultural and farming sectors across Africa, often with antibiotics of concern, adding selection pressure for AMR [30,31,32,33]. This is leading to increasing calls for a One Health approach across Africa, coupled with measures to prohibit the self-purchasing of antibiotics to reduce future AMR [34,35,36,37].

The costs associated with AMR are also considerable and will continue to rise unless addressed [38]. The World Bank (2017) recently estimated that even in a low-AMR scenario, the loss of world output due to AMR could exceed USD 1 trillion annually after 2030 [39]. This could potentially increase up to USD 3.4 trillion annually, equivalent to 3.8% of annual global Gross Domestic Product [39]. As a result, increasing AMR could be devastating, becoming the next pandemic unless considerable activities are undertaken to improve the future use of antibiotics [40].

Concerns with increasing morbidity, mortality and costs associated with AMR have resulted in a plethora of international, regional, national, and local activities and strategies to try and address the situation [41]. International activities include those among the Interagency Coordination Group on Antimicrobial Resistance (ICGAR) Global Leaders group, ongoing activities within the Organisation for Economic Co-operation and Development (OECD), the Fleming Fund, and the World Bank alongside activities by the World Health Organization (WHO) including the One Health Approach and the development of Global Action Plans (GAP) to tackle AMR [36,37,42,43,44,45,46,47,48]. Regional initiatives within Africa include activities by the Africa CDC, the African Society for Laboratory Medicine (ASLM), the Southern Africa Centre for Infectious Disease Surveillance, other Civil Society Organisations in Africa as well as the development of African guidelines to treat common bacterial infections across age groups [49,50,51,52,53,54].

Alongside this, the WHO has reclassified antibiotics into the AWaRe list (‘Access’, ‘Watch’ and ‘Reserve’) taking into account the impact of different antibiotics and classes on resistance potential to reduce future AMR [55,56]. The ‘Access’ group are considered as first- or second-line antibiotics for up to 26 common or severe clinical syndromes, typically with a narrow spectrum and low resistance potential. The ‘Watch’ group have a higher resistance potential and side-effects, with the ‘Reserve’ group only recommended as last resort antibiotics and prioritized for Antimicrobial Stewardship Programs (ASPs) using agreed quality indicators (QIs) (Figure 1) [55,56,57,58,59]. Assessing antimicrobial prescribing against current guidance, and monitoring their use based on the WHO AWaRe list, is increasingly being undertaken across Africa and beyond to improve prescribing given the extent of current prescribing of ‘Watch’ and ‘Reserve’ antibiotics in Africa [13,15,58,59,60,61,62,63,64,65,66,67].

The WHO GAP resulted in the development of National Action Plans (NAPs) across countries to combat rising levels of AMR [68,69]. Africa is no exception; currently, African countries are at different stages of implementation and monitoring of their NAPs [70,71,72,73,74]. A key activity within the NAPs is understanding current antimicrobial utilization and resistance patterns. Some African countries have high antimicrobial utilization rates within hospitals, with published studies across Africa reporting prevalence rates between 52.0% and 88.2% of in-patients [13,14,15,16,19,63,75,76,77,78,79,80,81,82]. There has also been considerable disquiet with high rates of extended antibiotic prophylaxis to prevent surgical site infections (SSIs) across Africa [83]. This is a concern as such practices may increase adverse reactions, AMR and costs with limited or no evidence on further reducing SSIs [83]. We are already seeing high rates of AMR among patients undergoing surgery who develop SSIs among patients in African hospitals, and this urgently needs to be reversed [84].

The use of antibiotics in patients with COVID-19 in hospitals has increased exponentially since the start of the pandemic. This is despite only a limited number of patients actually being diagnosed with bacterial or fungal infections reflecting current challenges with routine culture and sensitivity testing across Africa [14,76,85,86,87,88,89]. This misuse of antibiotics has been exacerbated by national guidelines in Africa which advocated the prescribing of a number of antibiotics in patients with COVID-19 [90]. The authors were concerned with this worrisome development with only a limited number of patients likely to have a bacterial co-infection [90], with others documenting similar concerns [88,91,92,93]. This mirrors similar activities regarding the purchasing of antibiotics in patients with COVID-19 without a prescription [94,95], and both situations urgently need to be addressed to avoid further pressures to increase AMR [88,96,97,98]. Alongside this, whilst the use of antibiotics has changed during the COVID-19 epidemic, and even during different waves of the epidemic over time in the same country [99], which needs addressing, there remains a lack of availability and access to a number of antibiotics in many LMICs [100,101]. While ‘Access’ antibiotics are placed in this group due in part to their availability in hard-to-reach areas, the availability of/access to other groups of antibiotics including ‘Watch’ and ‘Reserve’ antibiotics for use in hospitals, together with newly developed antibiotics, which may be necessary to use in areas of high resistance where antibiotic stewardship systems are available, is sorely missing and desperately needed [102]. We will be pressing for this in the future along with measures to increase COVID-19 vaccine equity. African Governments and others need to ensure that healthcare professionals (HCPs) can readily administer COVID-19 vaccines to reduce future hospitalisations [103,104]. Increasing vaccinations will reduce infection rates and subsequent consequences including hospitalisations, thereby reducing unnecessary prescribing of antimicrobials and associated AMR [96,97,98,105,106]. However, this requires comprehensive programs across Africa to address current high vaccine hesitancy [107,108,109].

A key activity to reduce AMR is the instigation of pertinent ASPs containing quality indicators [110,111]. Current indicators include adherence to prescribing guidance, which is increasingly being used to improve future antimicrobial prescribing across Africa and beyond given concerns with adherence to guidelines in practice along with rising AMR rates [112,113,114,115,116,117,118,119,120,121].

The role of quality indicators, whether as part of ASPs or separately, is also growing across countries, including African countries, to improve antibiotic prescribing in hospitals, including those based on the AWaRe classification [58,60,64,112,122,123]. This is because one way of measuring, monitoring and improving the quality of care is to use indicators, which are standardized measures of healthcare quality that make use of readily available, easy-to-use data independent of subjective judgement.

There are three main types of indicators used in healthcare (Figure 1) and there must be a clear a priori purpose for both developing, collecting and using indicator data across Africa.

**Figure 1 antibiotics-11-01824-f001:**
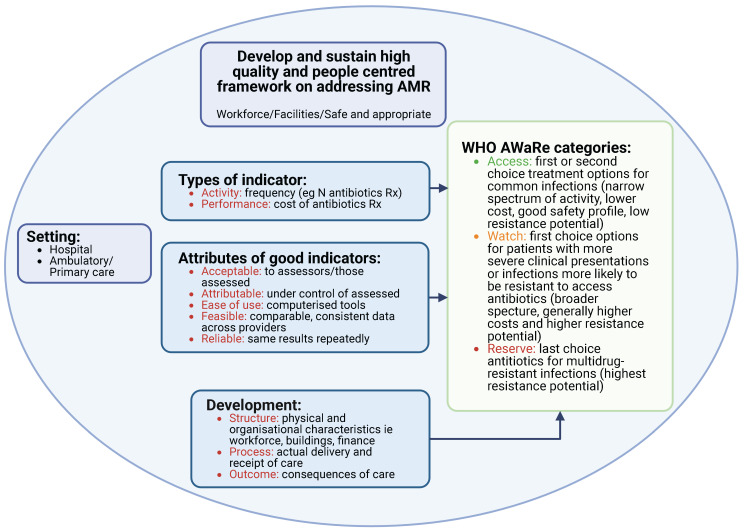
Key principles for indicator development for antimicrobials across Africa (adapted from [56,124,125,126,127]). NB: Created via Biorender (https://biorender.com/) with permission granted to Catrin Moore (accessed on 14 November 2022)).

However, there can be concerns instigating ASPs, with issues of resources, manpower, and knowledge among key stakeholders especially among low- and middle-income countries (LMICs) including African countries (Appendix A) [128,129,130,131]. This though is also starting to change among African countries as well as among other LMICs [111,132,133,134,135,136,137]. We are likely to see more ASPs being instigated across Africa and beyond as part of NAPs to reduce AMR, enhanced by the availability of toolkits from the WHO and others [67,70,132,137,138,139,140].

Consequently, this study aims to comprehensively document current antimicrobial utilization patterns in hospitals right across Africa from published point-prevalence surveys (PPS) with no discrimination based on geography and income status. Here we also concentrate on the extent of inappropriate prescribing of antibiotics post-operatively to prevent SSIs. It is important that antibiotic prophylaxis is administered for the appropriate duration to prevent SSIs as they are responsible for an appreciable number of hospital-acquired infections, impacting on morbidity, mortality and costs [141,142,143,144,145]. However, as mentioned, extended prophylaxis increases AMR and adverse reactions with limited or no impact on further reducing SSIs [83,141,146].

Alongside this, we will document potential prescribing and quality indicators that can be used across hospitals in Africa to improve future antibiotic prescribing based on published studies. In addition, document examples of successful ASPs that have been instigated across Africa to provide future guidance. We started with hospitals as more information is currently available regarding the prescribing of antibiotics in hospitals versus ambulatory care across Africa [70]. We will not discuss potential programs to address COVID-19 vaccine hesitancy across Africa as this has already been discussed [107,109]. Similarly, programs to reduce inappropriate use of antibiotics in agriculture and farming, with the emphasis on hospital in-patient prescribing. However, both are important under a One Health approach.

## 2. Results

The key issues we identified to improve future antimicrobial prescribing in hospitals across Africa are described below. These include current high utilization rates among a number of African countries where studies have been undertaken; post-operative prescribing for more than one day to prevent SSIs as well as concerns with the lack of adherence to guidelines and high empiric use of antibiotics. Several potential prescribing and quality indicators have been identified to improve future antimicrobial prescribing as part of ASPs. These are being utilized among a growing number of successful ASPs across Africa to improve future prescribing. Our results can provide exemplars going forward.

### 2.1. Current Antimicrobial Utilization Patterns in Hospitals across Africa

Appendix A documents current antimicrobial utilization patterns from 33 PPS studies that have recently been undertaken across Africa to a baseline for future ASPs. The documented studies include surveys from a single hospital within a country, with up to 18 hospitals in a country. However, Appendix A excludes the 12 African hospitals who were part of the Global PPS of Versporten et al. (2018) [112]. In addition, the 44 hospitals from sub-Saharan Africa that were part the updated Global PPS analysis, which includes the extent of Access, Watch and Reserve antibiotics prescribed [60]. These studies were excluded due to insufficient data for completion of Appendix A in the two publications [60,112]; the summary results though have been included. As seen, multiple PPS studies have been undertaken in low- and low-middle-income African countries despite concerns with available resources and personnel.

Antimicrobial utilization was typically high across most of the studied African countries (over 50% of patients), with similar rates in a recent study in Zambia [147]. Hospitals in Nigeria saw the highest antimicrobial utilization rates at 59.6–97.6% of in-patients surveyed Table 1. The lowest antimicrobial utilization rates were seen in South Africa (33.6–49.7% of patients). Incidentally, a similar range was seen among the 12 African hospitals taking part in the Global PPS of Versporten et al. (2018) [112].

There have also been studies assessing antimicrobial use among selected wards in hospitals in Uganda, with similar high utilization rates (79%) as seen among hospitals in Nigeria (Appendix A) [148].

The penicillin group (which are typically in the AWaRe ‘Access’ group) and the cephalosporins (typically third-generation cephalosporins, in the AWaRe ‘Watch’ category) were the most prescribed antibiotics among the African countries reporting their PPS results, followed by metronidazole. There was also high use of metronidazole (23.6%) and ceftriaxone (23.2%) and cefuroxime (8.3%) among the 44 hospitals from sub-Saharan Africa taking part in the Global PPS study of Pauwels et al. (2021) [60].

### 2.2. Antibiotic Prophylaxis to Prevent Surgical Site Infections

Table 1 documents the extent of extended antimicrobial prophylaxis within hospitals among African countries documenting their findings to prevent SSIs alongside details of the antimicrobials prescribed where documented. Antimicrobials prescribed for SAP were mainly third-generation cephalosporins including ceftriaxone (‘Watch’ antibiotic), metronidazole (‘Access’ antibiotic) and the penicillins including co-amoxiclav (typically ‘Access’ antibiotics). This is similar to the findings of Pauwels et al. (2021) who found that the most common antibiotics prescribed for SAP were ceftriaxone and metronidazole [60].

Published reasons for extended prophylaxis included resistance to change among HCPs, overcrowding in hospitals, concerns with hospital cleanliness, proper aseptic techniques not being followed during the operation, poor knowledge regarding antibiotics among physicians, concerns with malnutrition in some patients and patient expectations [114,149,150,151,152]. There can also be concerns with repeated door openings during surgery impacting on the potential development of SSIs [153].

Programs, including consistent and comparable ASPs, are urgently needed across Africa to reduce the extent of prolonged prescribing to reduce adverse reactions, AMR and costs [83]. Details of some of the interventions that have been successfully introduced across Africa to improve the prescribing of antibiotics to prevent SSIs, building on possible prescribing/quality indicators Table 2, are summarized in Table 3.

**Table 1 antibiotics-11-01824-t001:** Extent of prolonged antimicrobial prophylaxis to prevent SSIs across sub-Saharan Africa.

Country	Year (and Reference)	Findings
**Low Income ***		
Burkina Faso	2019 [154]	Prolonged administration of antibiotics was common (>2 days) among 62 patients in this PPS study to help prevent SSIs Antibiotics administered for >2 days in 87.1% of cases
Ethiopia	2018 [155]	Among patients who were prescribed antibiotics postoperatively (80 out of 90 patients) for SAP, 88.8% were prescribed for >1 day post-operatively The majority of patients (84.5%) received ceftriaxone (**W**)
	2018 [156]	79.1% of surveyed SAP patients (153 patients) were prescribed antibiotics for 2 days or more 34.7% were prescribed antibiotics for >5 days Approximately 84% of the patients were prescribed ceftriaxone (**W**) either alone or in combination
	2022 [157]	82.6% of 218 patients undergoing surgery had antibiotics prescribed for >1 day to prevent SSIs The average number of antibiotics prescribed per patient was 1.32 Ceftriaxone (54.7% of antibiotics—**W**) was the widely prescribed antibiotic for SAP
Rwanda	2019 [158]	Nearly all women who received antibiotics pre-operatively (66.7% of 550 women undergoing a caesarean section) had antibiotics post operatively to reduce SSIs Typically, a single dose of 1 g ceftriaxone (**W**) was given within 1 h before incision
Tanzania	2020 [159]	Out of 57 patients, 33% had antibiotics prescribed for SAP for 2–3 days and 56% for >3 days Ceftriaxone (**W**) was the most prescribed antibiotic for SAP (*n* = 28; 49%)
	2020 [160]	Typically, SAP was administered for 3 days post operatively prior to the intervention followed by oral antibiotics, i.e., 3 days of intravenous ceftriaxone (**W**) plus metronidazole (**A**), followed by oral penicillins (**A**) plus metronidazole for at least 5 days—timing was highly variable ranging from 1 to 24 h post-caesarean section This appreciably changed following the education of key stakeholders and monitoring of future prescribing habits with patients. A pre-operative prophylaxis with 1 g ampicillin (**A**) was administered 30–60 min before the incision with antibiotics only prescribed post-operatively for treatment Combined with training, overall a reduction in SSIs despite limiting antibiotic prescribing post-operatively among 320 studied patients Overall, the educational intervention provided savings of EUR 1500 for the hospital with reduced antibiotic prescribing
	2021 [81]	97% of patients undergoing surgery (96 patients overall) had antibiotics prescribed for SAP for >1 day
Uganda	2020 [161]	Most patients received prolonged antibiotic therapy for SAP after their surgery (907 patients) Combination of ceftriaxone (**W**) and metronidazole (**A**) was the most common regimen (609/907 patients—67.1%)
	2021 [81]	97.1% of antibiotics prescribed for SAP were for >1 day (170 patients overall)
	2022 [13]	98.4% of patients surveyed had multiple doses of antibiotics for SAP for >1 day (301 patients overall) Ceftriaxone (**W**) and metronidazole (**A**) were the principal antibiotics prescribed in this PPS study including for prophylaxis
**Low-Middle Income ***		
Congo	2020 [162]	69.1% of surveyed patients (265 patients in total) were prescribed antibiotics for 3 days or longer post operatively Ampicillin (**A**) was the most frequent antibiotic prescribed (43.8% of patients) followed by cloxacillin (13.2%) (**A**), gentamicin (9.4%) (**A**) and ceftriaxone (9.1%) (**W**) either alone or in combination
Ghana	2019 [163]	88.4% of patients (out of 121 patients) were prescribed antibiotics for >1 day to prevent SSIs 9.9% of patients were prescribed antibiotics for one day with only 1.6% of patients receiving a single dose of antibiotics to prevent SSIs The most commonly prescribed antibiotics for SAP were cephalosporins (28.9% of patients) and co-amoxiclav (**A**) (28.1%)
	2020 [76]	Antibiotics for SAP were typically prescribed for >1 day in 69.0% of patients in one of the surveyed hospitals and 77.0% in the other (26 patients overall)
	2021 [81]	75.5% of antibiotics prescribed for SAP were for >1 day (478 patients overall)
	2021 [164]	78.0% of 318 patients surveyed undergoing surgery received antibiotics for >1 day to prevent SSIs 13.0% of those surveyed were prescribed antibiotics for one day post operatively, and only 9% received a single dose
	2022 [141]	Duration of antibiotics to prevent SSIs among those surveyed was 6.9 ± 2.1 days The most common antibiotics prescribed were a combination of cefuroxime (**W**) and metronidazole (56.1%, *n* = 335) (**A**) followed by co-amoxiclav (14.2%) (**A**)
Kenya	2017 [165]	In all 69 patients surveyed, the duration of SAP ranged from one to three days Ceftriaxone (**W**) was the most common antibiotic prescribed for SAP (78% of patients)
	2018 [78]	The average number of doses of antibiotics for SAP in surveyed patients was 19.1 The most frequently prescribed antibiotics on the surgical wards (most for SAP) were third-generation cephalosporins
	2018 [166]	Most antibiotics prescribed in the surgical ward of this hospital were for prophylaxis (56.3%) vs. treatment (43.7%) The mean duration of antibiotic administration in the surgical ward was 6 ± 4.7 days, with most patients prescribed antibiotics for 1–3 days (30.2%) or 4–6 days (40.9%) The most prescribed antibiotics among those surveyed were ceftriaxone (**W**) and flucloxacillin (**A**) either alone or in combination with metronidazole (**A**)
	2019 [79]	76.9% of surveyed patients with antibiotics for SAP were prescribed these for >1 day with only 9.6% of patients administered a single dose for SAP
Nigeria	2016 [167]	Prolonged administration of antibiotics to prevent SSIs was common among 100 patients undergoing SAP along with the prescribing of broad-spectrum antibiotics including third-generation cephalosporins
	2017 [168]	Antibiotic prescriptions for SAP were given for >1 day in 95.0% of 277 patients undergoing surgery Ceftriaxone (28.0% of situations) (**W**), metronidazole (20.0%) (**A**) and cefuroxime (17.0%) (**W**) were the most prescribed antibiotics for SAP
	2020 [82,169]	All antibiotics prescribed for SAP were for >1 day in both studies In the study of Abubakar, nitroimidazoles including metronidazole (33.7% of total prescriptions), third-generation cephalosporins (20.8%), and combinations of penicillins and beta-lactamase inhibitors (10.9%) (**A**) were the most prescribed antibiotics
	2020 [114]	89% of surveyed patients (127 undergoing prophylaxis) were given antibiotics for >24 h to help prevent SSIs
	2021 [16]	94.8% of 96 patients had antibiotics administered for longer than 24 h to prevent SSIs 4.2% had antibiotics administered for one day and only 1.0% of patients were prescribed one dose of SAP Metronidazole (**A**), cefuroxime (**W**), ceftriaxone (**W**) and ciprofloxacin (**W**) were the most prescribed antibiotics in the surgical wards
	2022 [14]	All 63 patients were prescribed antibiotics for SAP post-operatively—24 h in 23.8% of patients and >1 day in 76.2% of patients
Zambia	2021 [81]	96.5% of patients undergoing surgery had antibiotics prescribed for SAP for >1 day (83 patients overall)
**Upper-Middle Income ***		
Botswana	2018 [170]	Prolonged administration of antibiotics was common for SAP with a mean (SD) duration of 5 (+/− 2.6) days, greatest for emergency surgery (72.7% of occasions among 104 patients) The most commonly prescribed antibiotics were cefotaxime (80.7% of situations) (**W**), metronidazole (63.5%) (**A**) and cefradine (**A**) (13.6%)
	2019 [19]	Extended prophylaxis for SAP (>1 day) was common—greatest among Primary hospitals (100% of 2 patients surveyed) and Tertiary hospitals (100% of 58 patients surveyed) versus District hospitals (90.3% of 31 patients surveyed) and Specialist hospitals (66.7% of 27 patients surveyed) Principal antibiotics prescribed for SAP were ampicillin (26.77% of occasions) (**A**), amoxicillin (24.41%) (**A**), metronidazole (17.32%) (**A**) and ceftriaxone (7.09%) (**W**)
South Africa	2021 [171]	In 73.2% of cases surveyed (*n* = 108 patients) for SAP were prescribed antibiotics for >1 day Cefazolin (**A**) was the most commonly prescribed antimicrobial (45.5% of cases) for SAP followed by co-amoxiclav (22.3% of cases) (**A**), ceftriaxone (9.8%) (**W**) metronidazole (5.4%) (**A**)
	2022 [61]	In 66.7% of paediatric cases (10 out of 15 patients), antibiotics for SAP were prescribed for >1 day

A and W: Access and Watch (AWaRe classification); SAP: Surgical Antibiotic Prophylaxis; SSIs: Surgical Site Infections; * World Bank Status (Based on [59]).

### 2.3. Prescribing Indicators Currently Being Used in Hospitals to Improve Antimicrobial Prescribing

A variety of prescribing indicators have been used among hospitals across Africa to enhance future prescribing as described in Table 2. These reflect increasing activities among hospitals across Africa to improve the future prescribing of antimicrobials in their countries thereby helping to reduce AMR.

However, a major concern across Africa is the current lack of electronic healthcare systems in hospitals to routinely track standards of care using consistent and comparable coding across countries. As a result, periodic surveys are typically undertaken to monitor care. This is likely to change as more applications and other electronic tools become available across Africa to track prescribing, building on current initiatives [61,62,172].

**Table 2 antibiotics-11-01824-t002:** Indicators that have been used among in-patients in hospitals across Africa to assess the prescribing of antibiotics.

Indicator	References
Activity/Performance Indicators	
% of in-patients prescribed antibiotics in a single PPS/ over specific time periods, e.g., successive waves of COVID-19	[19,61,112,172]
% of antibiotics prescribed by defined daily doses (DDDs), e.g., DDDs/1000 patient-days in a PPS or over a specified time	[171,173,174,175]
% of a course of antibiotics prescribed (duration) in accordance with agreed guidance/ Days of antibiotic therapy per 1000 patient-days	[166,176]
% of antibiotics administered to in-patients within the first hour of prescribing within a designated period of time	[177]
% of patients where the indication for prescribing and/ or stop and review dates are included in patients’ notes	[15,19,76,81,114,168,169,178,179]
% oral vs. IV antibiotics (including as part of de-escalation policies)	[15,76,82,114,166,168,171,178,179,180,181]
% of missed doses documented in patients’ notes, e.g., as part of a PPS	[19,148]
% of antibiotics prescribed by their international non-proprietary name, e.g., as part of a PPS	[182,183]
% compliance to agreed process measures surrounding AMS	[184]
% of patients prescribed antibiotics within the country’s essential medicine list over an agreed period of time	[61,171,180,182,183]
**Process quality indicators**	
% of in-patients prescribed antibiotics in adherence to agreed guidelines within a specified time period/part of a PPS	[81,112,134,168,184,185,186,187,188,189,190,191]
% of patients prescribed a course of antibiotics in accordance with guideline duration recommendations within a specified time period/ part of a PPS	[166,176]
% of patients where cultures are taken and sent for analysis to guide antibiotic prescribing/ targeted therapy within a specified time period/ part of a PPS	[76,114,169,192]
% of antibiotics prescribed based on the AWaRe classification/% reduction in the prescribing of target antibiotics, e.g., ‘Watch’ cephalosporins to potential ‘Access’ antibiotics (current target is 60% of current prescribing should be ‘Access’ antibiotics)	[60,76,81,193]
% of patients prescribed antibiotics post-operatively to prevent SSIs/% appropriate use of antibiotics to prevent SSIs during an agreed time period	[194,195]
% of key antibiotics available for prescribing/ Whether there are agreed therapeutic interchange policies in the hospital when there are likely to be shortages of standard antibiotics for the condition (over a specific time period)	[183,196]
% of all admitted patients with pneumonia to the hospital correctly classified and treated to agreed guidelines (over a specified time period)	[187,190]
**Outcome Indicators**	
% SSIs following operations (over an agreed time period)	[160,194,197]
% Mortality rates (post-intervention versus pre-intervention) following changes in antimicrobial prescribing, e.g., reducing extensive antimicrobial prescribing post-surgery for SAP or reducing extensive prescribing of ‘Watch’ antibiotics	[175,176,193]

AMS: Antimicrobial Stewardship; DDDs: Defined Daily Doses; SAP: Surgical Antibiotic prophylaxis; SSIs: surgical site infections.

### 2.4. Antimicrobial Stewardship Programs 

At least 20 ASPs have been successfully instigated across Africa in recent years (between 2013 and 2022) to improve antimicrobial prescribing, with checklists and guidance now being developed for sub-Saharan Africa and beyond to help with their development and implementation [135,139,198]. Box 1 contains key areas for hospitals to concentrate on when seeking to introduce sustainable programs to improve future antimicrobial use in hospitals.

Box 1Selected Antimicrobial Stewardship Checklist (adapted from [135]).Has your hospital management formally identified AMS as a priority objective and included it as a key performance indicator?Does your hospital have a formalized structure and group responsible for AMS activities including researching and promoting appropriate antibiotic use as part of agreed ASPs?Is this currently a multidisciplinary AMS group available in your hospital to implement agreed ASPs, and does this group include a designated leader?Is there access to HCPs in infection management and stewardship in the hospital willing to be part of AMS teams?Does your hospital currently offer educational resources to support training of HCPs regarding antimicrobial prescribing and its monitoring to improve future care?Is there dedicated and sufficient budget to support AMS activitiesDo you have access to laboratory/imaging services to support improved antibiotic use and away from untargeted and unnecessary prescribing, and are the results available in a timely manner to support diagnosis and appropriate antibiotic prescribing?Does your ASP currently monitor compliance with one or more agreed interventions, e.g., improved compliance to national or local guidelines, and report back the findings to improve future care including any changes in the quality/ appropriateness of antimicrobial prescribing in agreed areas?Has your hospital conducted a PPS in the past year and used the findings to improve future antimicrobial prescribing?Does your hospital have available and up-to-date recommendations for infection management, and are these readily available to prescribers?Does your hospital currently have any published AMS protocols such as a restricted antimicrobial list especially surrounding ‘Watch’ and ‘Reserve’ antibiotics and IV to oral switching policiesDoes your hospital currently have any published Infection Prevention and Control protocols, and are these regularly monitored, e.g., surrounding hand hygiene protocols?
AMS—Antimicrobial Stewardship; ASP—Antimicrobial Stewardship program; HCP—Healthcare Professional.

Their details and documented impact are summarized in Table 3. Typically, multiple interventions are more successful with improving future prescribing than single interventions. In addition, activities need to be continually followed up. Otherwise, prescribing habits could drift back towards pre-intervention levels.

One problem is the regular turn-around of junior staff, who subsequently need training on antimicrobial prescribing for continuous improvement. The increasing availability of electronic tools should help in this regard in the future [199].

**Table 3 antibiotics-11-01824-t003:** Summary of published studies across Africa documenting ASPs and their impact.

Author, Country and Year	Intervention and Aim	Impact of the Intervention
**Low Income ***		
Gebretekle et al., Ethiopia, 2020 [176]	1109 individual patients took part (707 during the intervention and 402 in the post-intervention periods) Principally Education as an intervention. This included: Intervention—weekly audit meetings and immediate (verbal and written) feedback sessions regarding antibiotic prescriptions of admitted patients on 4 wards This built on recently developed institutional guidelines and training sessions with relevant clinicians on ASPs and guidelines Aim: auditing of antibiotic prescriptions post intervention However, no feedback initiatives to remind physicians	Most commonly prescribed antibiotics were ceftriaxone, cefepime, meropenem, metronidazole and vancomycin 96% of the recommendations made by the AMS team were accepted Once the intervention ceased, total antimicrobial use increased by 51.6% and the mean duration of treatment increased by 4.1 days/patient respectively Mean hospital stay and crude mortality decreased during the intervention; however, increased significantly after the intervention
Alabi et al., Liberia, 2022 [134]	Intervention: education and engineering involving a collation of three activities: ○ production and dissemination of local treatment guidelines ○ training and regular AMS ward rounds ○ monitoring agreed QIs QIs included prescribing of correct antibiotics (incorporating completeness of microbiological diagnostics) as well as dosages and duration QIs were assessed in a case series after AMS ward rounds and fed back to key personnel 620 patients overall—310 pre intervention and 310 post intervention Aim: Assess the impact of AMS programs with improving antibiotic prescribing	Improvements were seen in all QIs: Adherence to local guidelines improved from 34.5% (107/310) to 61.0% (189/310) (*p* < 0.0005) Correct dosing improved from 15.2% (47/310) to 36.5% (113/310) (*p* < 0.0005) Optimal duration of antibiotic use improved from 13.2% (41/310) to 31.0% (96/310) (*p* < 0.0005) Proportion of patients receiving ceftriaxone reduced from 51.3% (159/310) to 14.2% (44/310) (*p* < 0.0005). Following the ASP, 79.7% (247/310) of patients had samples sent for microbiological analysis
Lester et al., Malawi, 2020 [193]	Intervention: education and engineering involving guidelines, posters and the application of smartphones to help with clinical decision making as well as regular PPS studies combined with prescriber feedback 503 patients were involved—203 pre implementation, 200 in the implementation phase and 100 patients post implementation Aim: Reduce extensive prescribing of third-generation cephalosporins within the hospital and associated costs with no adverse impact on mortality—especially with high rates of HIV among in-patients in the hospital (approximately 61% across the surveys)	The proportion of prescriptions for an IV 3rd-generation cephalosporin fell from 80.1% (193/241) of all prescriptions in the first survey to 53.6% (177/330) by the last survey The median length of a ceftriaxone course was reduced from 5 to 4 days aided by an increase in the number of clinician reviews of prescriptions at 48 h—increasing from 22.4% (54/241) at the start to 73.3% (242/330) by the final antibiotic survey Overall annual savings from the 3 wards was estimated at USD 15,000 with no change in mortality or median length of hospital stay
Suliman et al., Sudan, 2020 [188]	Intervention—principally education. Activities included: ○ Verbal contact by clinical pharmacists with all consultants and registrars involved with performing emergency caesarean sections (ECSs) separately about agreed updated guidelines for the use of prophylactic antibiotics in ECS to prevent SSIs ○ Brochures giving details about proposed changes in prophylactic antibiotic recommendations for patients undergoing ECS ○ These included no longer administering metronidazole (IV before cord clamping and on discharge) and oral amoxicillin-clavulanic acid on discharge ○ Subsequent auditing and feedback of the findings Overall, 195 participants were included, 94 participants before and 101 participants after the intervention Aim: To improve the rational use of prophylactic antibiotics among patients undergoing a cesarean section and to assess the impact of clinical pharmacist intervention on subsequent antibiotic utilization/ adherence to guidelines and possible cost-savings	The hospital protocol was fully followed so no patient subsequently received either metronidazole (IV or oral) or oral amoxicillin-clavulanic acid on discharge following the ASP intervention Cost saving of 31% on antibiotics administered to prevent SSIs post ASP No patient in the revised administration group developed any symptoms or signs of SSI (at days 15 and 30 post discharge)
Gentilotti et al., Tanzania, 2020 [160]	Intervention: principally Education. Activities included formal and on-job training including seminars on infection prevention and control/ evidence-based education on antimicrobial resistance and good antimicrobial prescribing practice Prior to this—antibiotics were typically prescribed post-operatively (98.2%) and for 8–10 days when given 1377 women undergoing caesarean sections were enrolled, 664 in the pre-intervention phase and 713 in the post-intervention phase Aim: Enhance appropriate antibiotic prescribing to prevent SSIs for patients undergoing caesarean sections	Pre-incision antibiotic prophylaxis was administered in significantly more cases post the educational intervention (*p* < 0.001) The extent of antibiotics administered post-operatively to prevent SSIs was also appreciably lower post intervention (*p* < 0.001) The timing of prophylaxis was adequate only in 28% of cases in the post intervention group, but this did not seem to affect SSI prevalence rates The total number of SSIs decreased from 48% pre-intervention to 17% post intervention (*p* < 0.001)
Ashiru-Oredope et al., 2022 [135]	Intervention: Principally Education and Engineering. Activities included developing a checklist of 54 items across 8 sections to identify current AMS activities surrounding key areas (highlighted in Box 1) across 19 participating hospitals with the number of inpatient beds ranging from 100 to 2000 (average 536) Educational initiatives undertaken to improve AMS capabilities within the hospitals, which included guideline development and promotion Post-intervention monitoring to record improvements in AMS activities to improve future antimicrobial prescribing	Improvements in AMS activities recorded across all hospitals (overall 79% improvement in AMS activities)—before this program only 3 hospital sites had a formal AMS structure Increased multidisciplinary membership of AMS teams including an increasing number of nurses and pharmacists to assist with future sustainability of AMS activities New guidelines, policies, posters developed/ implemented across participating hospitals to improve future antimicrobial prescribing Increased awareness of the WHO AWaRe classification of antimicrobials across most participating hospitals (79%)
Ngonzi et al., Uganda, 2021 [197]	Intervention: Principally Education and Engineering regarding the World Health Organization’s checklist of activities to reduce SSIs in patients undergoing caesarean sections Educational interventions combined with daily audits and feedback 678 patients’ charts were reviewed (200 in the pre-intervention phase, 230 in the intervention phase and 248 in the post-intervention phase). Aim: reduction in SSIs among patients undergoing caesarean sections in the hospital	The use of the WHO’s checklist for SSIs increased from 7% (13/200) pre-intervention to 92% (211/230) in the intervention phase (*p* < 0.001) Subsequently, fell to 77% ((191/248) post-intervention (*p* < 0.001) Prescribing of antibiotics rose from 18% (36/200) of patients pre-intervention to 90% (208/230) in the intervention phase (*p* < 0.001); subsequently, reduced to 84% (208/248) post-intervention phase (*p* < 0.001) The documented SSI rate fell from 15% pre-intervention phase to 7% in the intervention phase (*p* = 0.02)
**Low-Middle Income ***		
Aitken et al., Kenya, 2013 [152]	Intervention: Education and Engineering to develop, implement and monitor a policy within the hospital to improve post-operative prescribing of antibiotics among patients undergoing surgical operations Aim: Improve antibiotic prescribing for SAP and reduce costs	Appreciable improvement in reducing extensive post-operative prescribing of antibiotics to 40% (18/45) of operations within the first week and just 10% (5/50) by week 6 following the policy implementation (*p* < 0.0001) Overall, net reduction in the costs for IV antibiotics and associated consumables used to prevent SSIs at approximately, USD 2.50/operation
Amdany et al., Kenya, 2014 [181]	Intervention: Principally an educational initiative to enhance the use of oral vs. IV metronidazole including education, audit and feedback. Aim: Increase the use of oral vs. IV metronidazole	Post implementation audit showed an increase of more than 40% compliance in all the four criteria utilized to assess an increase in oral use. These are: ○ Criterion 1: Oral metronidazole is used in preference to IV metronidazole ○ Criterion 2: For each IV administration of metronidazole, there are records indicating why this route was used in preference to oral metronidazole ○ Criterion 3: For each IV administration, there are records indicating that the need was re-examined daily ○ Criterion 4: For each prescription for IV metronidazole, there are records indicating the switch to oral after significant improvement in patient’s condition and patients able to tolerate oral medication. As a result, reduced costs, patient discomfort and possible iatrogenic infections
Ntumba et al., Kenya, 2015 [194]	Intervention: Education and Engineering to improve the use of antibiotics in to prevent SSIs including reducing the number of patients prescribed antibiotics post-operatively. Activities included: Local adaptation of published guidelines Creation and tools for advocacy, training, and leadership around appropriate antibiotic use to prevent SSIs 406 patients pre-intervention, 353 post-intervention Aim: Improve antimicrobial use of SAP and reduce SSI rates	Patients receiving antibiotics post-operatively decreased from 50% to 26% Alongside this, crude SSI rates significantly decreased from 9.3% to 5% of patients
Ayieko et al., Kenya, 2019 [187]	Intervention—education and engineering involving two groups, with both groups receiving a half-day training on the new Kenyan pneumonia guidelines, with physicians in all hospitals supplied with updated protocol booklets including specific pneumonia algorithms. All hospitals also received continued network support The two groups were: (i) standard feedback with regular auditing and bimonthly feedback of general paediatric care and (ii) enhanced feedback group—Regular auditing of agreed indicators of pneumonia care, with monthly feedback using specific feedback sheets Overall 2299 childhood pneumonia admissions, 1087 within the hospitals randomized to enhanced feedback and 1212 to standard feedback Aim: Examined whether providing enhanced audit and feedback might accelerate adoption of new pneumonia guidelines	An improvement was seen in the enhanced feedback group regarding the correct classification and treatment of pneumonia after each round of enhanced feedback However, the performance declined in the standard feedback arm over time, which was attributable to consistently poor performances among four out of the six participating facilities
Allegranzi et al., Kenya, Uganda, Zambia, and Zimbabwe, 2018 [195]	Intervention: Education and Engineering to improve antibiotic prescribing for the prevention of SSIs Activities included: Five planned visits to each participating hospital among four African countries during the study period—supported by a range of educational tools Local teams identified key areas of concern with preventing SSIs; subsequently monitoring an agreed range of indicators (six pre-identified ones including skin preparation and optimal timing of prophylaxis) Subsequent launch of pertinent tools and agreed indicators alongside monitoring/feedback to improve future prescribing Aim: Improve antibiotic prescribing for the prevention of SSIs	Appropriate use of antibiotics to prevent SSIs improved from 12.8% (205/1604) at baseline to 39.1% (714/1827) in the follow-up phase (*p* < 0.0001) among the studied hospitals Concurrently, the cumulative incidence of SSIs decreased from a baseline of 8.0% (129/1604) to 3.8% (70/1827) post intervention (*p* < 0.0001)
Abubakar et al., Nigeria, 2019 [200]	Intervention: Principally education and engineering Activities included: The development and dissemination of an agreed protocol—agreed before its adoption to enhance subsequent adoption rates Educational meetings held with key clinicians to enhance the uptake of agreed protocols combined with wall mounted posters Alongside this, regular audit and feedback meetings using the baseline data to try and improve future antibiotic prescribing There were 226 and 238 surgical procedures in the pre- and post-intervention periods respectively Aim: To improve antibiotic prescribing by reducing the extent of extended prophylaxis to prevent SSIs.	Patients in the post-intervention period were 5.6 times more likely to receive antibiotics within 60 min before the incision to prevent SSIs vs. pre-intervention (*p* < 0.001) The prescribing of 3rd-generation cephalosporins for SAP was reduced from 29.2% in the pre-intervention period to 20.6% in the post-intervention period (*p* = 0.032). The rate of redundant antibiotic prescriptions was reduced by 19.1%—from 70.8% in the pre-intervention period to 51.7% in the post-intervention period The mean cost of SAP among patients was reduced by USD 4.2 (*p* < 0.001) after the interventions
**Upper-Middle Income ***		
Messina et al., South Africa, 2015 [177]	Intervention: Education and Engineering with pharmacists conducting daily AMS rounds in ICUs and ICU step-down wards among 33 private hospitals in South Africa to evaluate hang-time compliance among patients A total of 32,985 patients who received day 1 IV antibiotics were assessed for hang-time compliance Hang-time compliance was seen as patients receiving appropriate antimicrobials within an hour following the prescription Aim: To evaluate the change in compliance with administering antimicrobials within an hour of the prescription after implementation of a national antibiotic stewardship pharmacist-driven hang-time process improvement protocol	Overall hang-time compliance improved from 41.2% (164/398) pre-intervention to 78.4% (480/612) post-intervention (*p* < 0.0001) Post-intervention was analysed at week 60 among participating hospitals (*p* < 0.0001)
Brink et al., South Africa, 2016 [174]	Intervention: Principally Education. Activities involved: Initial training sessions with key stakeholders in each hospital among a total of 47 hospitals discussing the five process measures that would subsequently be audited by pharmacists in each hospital were provided through face-to-face regional learning sessions Subsequently, each pharmacist was required to undertake audits of the five measures in their hospitals The five measures included: (a) Cultures not performed before starting empiric treatment; (b) prolonged treatment (7 and 14 days); (c) more than 4 antibiotics prescribed concurrently; (d) concurrent double or EUR redundant antibiotic coverage 16,662 patients on antibiotics were reviewed during the 104 weeks of standardized measurement, with 7934 interventions by pharmacists recorded for the five targeted measures Aim: Improve antibiotic prescribing including increasing culture and sensitivity testing and reducing prolonged administration	Combined reduction in mean antibiotic prescribing in defined daily doses/ 100 patient days—down from 101.38 to 83.04 (*p* < 0.0001) Reductions across participating hospitals in the number of cultures not performed before starting empiric treatment or prolonged antibiotic treatment (7 and 14 days) Reductions also in the prescribing of more than 4 antibiotics prescribed concurrently, and the prescribing of concurrent double or redundant antibiotic coverage among participating hospitals
Boyles et al., South Africa, 2017 [175]	Intervention: Education and Engineering to improve future antibiotic use in the hospital Key activities included: 1. A comprehensive ASP program comprising online education, a dedicated antibiotic prescription chart and weekly dedicated ward rounds to discuss current prescribing practices—continued over 4 years 2. Pre- and post-intervention data compared to provide future guidance Aim: To improve future antibiotic use in the hospital	Total antibiotic consumption fell from 1046 defined daily doses/1000 patient days (pre-intervention) to 868 (first 2 years of the intervention—remaining at similar levels for the next 2 years). Improvements driven by reductions in IV antibiotic use, particularly ceftriaxone Laboratory testing increased over the same period Cost savings on antibiotics (inflation adjusted) were ZAR3.2 million over 4 years No significant change in mortality or 30-day readmission rates over the 4 years
Brink et al., South Africa, 2017 [189]	Intervention: Education and Engineering to reduce inappropriate prescribing of antibiotics to prevent SSIs. Key activities (driven by hospital pharmacists) included: Testing and revising the developed guidelines and toolkits at pilot sites prior to their launch at regional training and institutional workshops Obtaining consensus and endorsement from key professionals within each hospital—enhanced by adapting and modifying guidelines where appropriate (building on current knowledge within each participating hospital including current SSI rates) Choosing at least one or more surgical procedures to audit—including recording pre-intervention practices and trends to demonstrate improvements Measuring compliance to agreed measures over a 4-week period and giving feedback 24,206 surgical cases were reviewed during the 70 weeks of standard measurements Aim: Implement a model utilizing existing resources in order to improve antimicrobial use for SAP in line with current guidelines among 34 hospitals in South Africa	Significant improvement in compliance with all process measures (composite compliance—choice, dosage, timely administration and duration) from 66.8% to 83.3% (95% CI 80.8–85.8) SSI rate decreased by 19.7% from a mean group rate of 2.46 pre-intervention to 1.97 post-intervention (*p* = 0.0029) Timely administration of antibiotics increased to 56.4% of surgical patients (*p* < 0.0001)—representing a 62.4% increase Antibiotic choice consistent with the guidelines increased to 95.9% of patients and the duration of prophylaxis was now appropriate among 93.9% of patients
Junaid et al., South Africa, 2018 [192]	Intervention: Principally education in a single hospital over 3 years Key activities included: Weekly dedicated AMS ward rounds A dedicated prescription chart with key issues including dose, frequency, duration, route of administration and possible de-escalation Audit tools for pharmacy, IPC and ward rounds, with regular multi-professional patient reviews A hospital-wide education program incorporating current principles of AMS, posters and e-training modules Infection prevention and control program monitoring Aim: Describe the development of an institutional ASP over a 3-year period in a single hospital and its impact to provide future guidance	Dosing considerations completed on patient’s charts improved for weight and eGFR; however, allergy entries decreased leading to additional training Education on sending of cultures prior to antibiotic commencement resulted in increased awareness of HCPs need to improve future prescribing Staff members reported increased knowledge on AMS principles following the various ASP activities
van den Bergh et al., South Africa, 2020 [184]	Intervention: Principally education to improve compliance to agreed guidelines for CAP. Activities included: A CAP bundle was developed which incorporated seven process measures, which included admission criteria, antibiotic choices, dose and length, as well as three outcome measures including length of hospital stay and mortality, which pharmacists subsequently used to audit compliance to the bundle and provide feedback Training sessions were conducted on the CAP guidelines and implementing ASPs within hospitals across South Africa. Following each learning session, a checklist of essential activities and deadlines was provided to each attending pharmacist Baseline data were collected to identify areas for improvement In a four-week period following the learning sessions, pharmacists subsequently applied the learnt ideas to improve compliance to the CAP guidelines and ways to give feedback to address identified gaps to further improve future compliance Overall, 3117 patients were reviewed of which 2464 were included in the final analysis—1247 patients at baseline that were compared to 1217 post intervention. Aim: To improve compliance to agreed guidelines for CAP to improve future care of patients	2464 patients from 39 hospitals were included with the ASP showing positive results: CAP bundle compliance improved from 47.8% to 53.6% (*p* < 0.0001) Diagnostic stewardship compliance improved from 49.1% to 54.6% (*p* < 0.0001) Improved compliance with process measures was significant for 5 of the 7 components, which included choice and dose of antibiotics prescribed as well as IV to oral switching However, there was no significant difference in mortality or median length of stay pre- and post-intervention
Bashar et al., South Africa, 2021 [173]	Intervention: Education and Engineering involving regular ASP ward rounds on two surgical wards During the ward rounds—each condition was discussed especially concerning antibiotic selection and laboratory investigations In addition, potential switching from intravenous to oral agents, dose optimisation and any dose adjustments in patients with renal and hepatic impairment 476 patients were involved—264 at baseline vs 212 in ASP phase Aim: Demonstrate a reduction in antibiotic usage (measured by the volume of antibiotic consumption following the ASP)—as a result improve overall antibiotic prescribing	Reduction in the volume of antibiotic consumption from 739.30 DDDs/1000 to 564.93 DDDs/1000 patient days following the ASP Reduction in inappropriate antibiotic use from 35% to 26% of patients An overall increase in culture targeted therapy Reduction in antibiotic administration for more than one day post operatively to prevent SSIs (from 7.3% to 6.6%) Small (non-significant reduction) in total antibiotics administered IV (from 89.4% to 84.2%) alongside an increase in appropriate IV administration from 56.9% to 60.8%

AMS: Antimicrobial Stewardship; ASP: Antimicrobial Stewardship Program; CAP: Community Acquired Pneumonia; ECS: Emergency Caesarean Sections; ICU: Intensive Care Unit; HCP: Healthcare Professional; IPC: Infection, Prevention and Control; PPS: Point Prevalence Survey; QI: Quality Indicator; SSI: Surgical Site Infection; * World Bank Status.

### 2.5. Suggested Activities to Improve Future Antimicrobial Prescribing in Hospitals

The suggested strategies to improve antibiotic utilization among hospitals across Africa, which will be crucial to reduce AMR, have been divided into short- and long-term initiatives (Table 4). These build on the potential for developing and expanding the use of digital technologies surrounding electronic health records and electronic prescribing to improve appropriate antibiotic use [199,201,202,203]. Such developments are essential to develop and regularly monitor antimicrobial prescribing against agreed prescribing/quality indicators.

## 3. Discussion

Reducing the burden of AMR is a high priority across Africa given its appreciable impact on morbidity, mortality and costs [38,39,43,232]. This is in part driven by the inappropriate and overuse of antimicrobials; however, this association appears less clear cut in LMICs due to the risks of contagion [28,232,233,234,235,236,237]. The multiple PPS studies undertaken across Africa in recent years (Table 1) have shown considerable usage across most African countries compared with other countries and continents [112], with the highest rates seen in Nigeria between 59.6 and 97.6% of surveyed patients. These utilization rates are appreciably higher than the suggested WHO target of 40% of hospital in-patients [238]. The lower utilization rates seen in South Africa, at 33.6% to 49.7% of hospital in-patients, may reflect the fact that the South African Government launched its ‘Antimicrobial Resistance National Strategy Framework’ in 2014, coupled with the availability of microbiology laboratories and the performance of hospitals being regularly monitored since 2014 [171,239]. Greater implementation of the NAPs is needed to reduce future utlization rates in hospitals; however, some African countries have only just started on this activity [70]. An average compliance of 59.5% to the National strategy was recently recorded among 26 public sector facilities across South Africa helping to improve antimicrobial prescribing and reducing AMR [239].

Encouragingly, the penicillins (typically in the Access group) and metronidazole were among the most prescribed antibiotics across Africa (Appendix A), with currently limited prescribing of ‘Reserve’ antibiotics. However, there was appreciable prescribing of cephalosporins, which are typically third-generation cephalosporins incorporating ceftriaxone, in hospitals including for SAP, which is a concern as ceftriaxone is a ‘Watch’ category antibiotic. Greater prescribing of ‘Access’ antibiotics in hospitals, where appropriate, can be achieved through establishing pertinent prescribing/ quality targets as well as monitoring subsequent utilization patterns in hospitals as part of ASPs. This can be part of a plethora of both short- and longer-term initiatives that can be undertaken across Africa to improve future antimicrobial use and combat AMR (Table 4). Other prescribing/quality targets are needed to ensure appropriate use of SAP to prevent SSIs by moving away from extended use post-operatively. As seen (Table 1), there is currently appreciable extended use of antibiotics for SAP across Africa, which needs urgently to be addressed to reduce adverse reactions, AMR and costs [83]. Successful ASPs have been implemented among African countries to improve antimicrobial use for SAP (Table 3) providing exemplars.

The number of PPS studies have grown across Africa over time (Appendix A) despite concerns with available resources and personnel, providing future guidance, and this acceleration will continue. In addition, we are seeing the number of successful ASPs increase across Africa (Table 3), despite again initial concerns regarding available financial resources and personnel to conduct ASPs in LMICs, providing exemplars to others [128,132,137,138]. This will continue as part of NAPs to reduce rising AMR rates across Africa [2,70]. 

Potential strategies for all key stakeholders to improve future prescribing of antimicrobials in the hospital sector have been consolidated into suggested short-, medium- and longer-term activities to provide future direction (Table 4). The key is Government commitment and activities through NAPs, and we are already seeing African countries develop and implement these [70]. However, considerable challenges still remain in terms of available finances to undertake agreed activities as well as available personnel to undertake suggested ASP activities and monitor their progress in hospitals. We will continue to monitor the situation given, as mentioned, concerns with rising AMR rates across Africa and the resultant impact on mortality and costs.

It is imperative that Africa progresses with activities to reduce AMR, with AMR seen as the next pandemic and the highest resistance rates are currently in Africa [2,40]. Suggested future research activities will also include a greater understanding of current antimicrobial utilization patterns in ambulatory care given the extent of utilization in this sector versus hospital use, especially for self-limiting conditions such as acute respiratory tract infections [12,70]. Increased digitalization of patient records within healthcare systems across Africa will assist with this [240].

We are aware of a number of limitations with this paper. These include the fact that we have not undertaken full systematic reviews for each topic including PPS and SSI studies as well as QIs and ASPs for the reasons discussed. However, we have documented an appreciable number of PPS and SSI studies across Africa, together with current prescribing/ quality indicators in use and ASPs. This has been achieved with the considerable knowledge of the senior level co-authors, similar to discussions on potential future strategies. Despite these limitations, we believe our findings and suggestions are robust given the extent of examples coupled with our methodology providing future direction.

## 4. Materials and Methods

The principal approach was a narrative review of key areas. This was supplemented by the considerable experience of the co-authors across countries and continents dealing with patients with infectious diseases as well as recording current utilization patterns, implementing policies to improve future prescribing including the development of pertinent quality indicators as well as researching and implementing ASPs.

This mirrors similar studies undertaken by the co-authors across a number of African countries and wider when providing future guidance regarding the management of both infectious and non-infectious diseases, as well as more general approaches, and is in line with institutional guidance [9,12,70,83,95,107,221,241,242,243,244,245,246].

### 4.1. Antimicrobial Utilization Patterns in Hospitals across Africa

The methodology built on a recent systematic review of PPS studies undertaken by some of the co-authors [247], and involved studies from 2016 onwards until October 2022. This methodology was employed since some of the sourced studies known to the co-authors would not have been incorporated in databases including PubMed and Web of Science. In addition, the principal objective of this paper was to document the findings from across Africa to provide a basis for the future. As such, we did not pre-specify which African countries would be included in this narrative review in order to provide as complete a picture as possible to provide exemplars for the future.

Similar to the systematic review of Saleem et al. (2020) [247], key categories included the number of participating hospitals, the PPS methodology, e.g., ECDC, Global PPS or WHO [16,65,77,81,238]; first, second or third most prescribed antibiotic broken down by ATC code and AWaRe classification [56,58,248]; whether prescribed for prophylaxis or treatment and the average number of antibiotics prescribed per patient.

As mentioned, we did not include the 12 hospitals taking part in the Global PPS study of Versporten et al. (2018) in the collation of published PPS studies alongside the African hospitals taking part in the study of Pauwels et al. (2021) as different parameters were collected in these studies including details of the most prescribed antibiotics across the indications [60,112]. However, we did include the study of D’Arcy et al. (2021) involving several African countries as this did contain relevant detailed information [81].

We are aware that some of the PPS studies referenced may contain the same hospital. For this reason, we did not include in the Table (Appendix A) the total number of hospitals per country in the various PPS studies. The intention was to list the various studies as exemplars going forward. The various African countries were broken down by their World Bank classification, i.e., low-income, low-middle and upper-middle-income countries, building on the recent study of Adekoya et al. (2021) for consistency [59]. This is because, as mentioned, there have been concerns with available resources and personnel within hospitals among LMICs to undertake PPS studies, and we wanted to explore this further.

### 4.2. Antibiotic Prophylaxis to Prevent Surgical Site Infections

The principal approach was a narrative review, building on recent publications involving some of the co-authors [83,150]. This was supplemented by additional studies from 2016 onwards known to the co-authors, which included details of antibiotics being prescribed to prevent SSIs incorporated in the sourced PPS studies (Appendix A). This is similar to the approach adopted by the co-authors in other studies. The various African countries were again broken down by their World Bank classification, i.e., low-income, low-middle and upper-middle-income countries, building on the recent study of Adekoya et al. (2021) for consistency [59].

### 4.3. Prescribing Indicators

The principal approach was again a narrative review building on recent publications involving the co-authors supplemented by additional studies known to the senior-level co-authors. This is similar to the approach adopted by the co-authors for the PPS and SSI studies.

### 4.4. Antimicrobial Stewardship Programs

The principal approach was a narrative review of recent ASPs that had been instigated across Africa. This built on recent reviews coupled with additional studies known to the co-authors from 2013 onwards [132,137,138]. The objective was again to provide guidance to African countries planning ASPs rather than undertaking a systematic review of the studies.

In order to enhance understanding, the different activities that can be undertaken by groups within hospitals when instigating ASPs will be broken down into the 4Es. These are Education, Engineering, Economics and Enforcement [83,249]. Education incorporates a number of activities including developing and communicating formularies as well as developing and monitoring adherence to agreed guidance [116,214,215,249,250]. Engineering includes organizational or managerial interventions [240]. This incorporates for instance prescribing targets such as an agreed percentage of antibiotics being prescribed according to accepted guidelines, an agreed percentage of prescribing of ‘Access’ antibiotics from the WHO AWaRe list, as well as an agreed percentage of patients prescribed short courses of antibiotics to prevent SSIs [9,58,116]. Economics includes financial incentives to clinicians, patients, pharmacists or hospitals to improve the rational use of medicines such as incentives for clinicians when reaching agreed prescribing targets as well as fining pharmacists for dispensing an antibiotic without a prescription when this is prohibited [249,251]. Enforcement includes enforcing regulations by law including prohibiting the dispensing of antibiotics within pharmacies without a prescription or regulations banning the use of colistin unless under strict regulations [9,251,252,253].

The various African countries were again broken down by their World Bank classification, i.e., low-income, low-middle and upper-middle-income countries, building on the recent study of Adekoya et al. (2021) for consistency [59]. This is because, as mentioned, there have been serious concerns about the ability of especially low- and low-middle-income countries to undertake ASPs in practice due to lack of resources and personnel [128].

## 5. Conclusions

In conclusion, reducing AMR has to be a high priority among all African countries given its clinical and economic impact. Without such activities, AMR will become the next pandemic. However, reducing AMR rates requires multiple coordinated activities across sectors driven by Governments and others across Africa as part of NAPs. This includes an urgent need for HCPs to appreciably reduce inappropriate prescribing of antibiotics across hospitals as well as increased cognisance of classifications and their implications such as the AWaRe classification when prescribing. This necessitates active surveillance of current utilization and resistance patterns across hospitals as well as initiating ASPs for target areas. Such activities include reducing the extent of antibiotic prophylaxis post-operatively for SAP, routinely incorporating the rationale for prescribing of antibiotics in patients’ notes alongside inserting start and stop dates, as well as developing and disseminating locally agreed guidelines.

This is essential given limited new antimicrobials being developed as well as concerns with the routine availability of specific antibiotics to tackle resistance; however, this is compensated, to some extent, by developments in vaccine technologies. The latter will require strategies to address the current high rates of vaccine hesitancy that exist across Africa as seen in the recent COVID-19 pandemic. A coordinated approach including all key stakeholder groups is also essential to minimize misinformation and maximize the impact of future interventions to reduce AMR rates. This can be part of an agreed One Health approach incorporated into NAPs.

## Figures and Tables

**Table 4 antibiotics-11-01824-t004:** Suggested strategies to improve future antimicrobial utilization among hospitals across Africa.

Timescale	Potential Strategies
**Short to Medium Term (e.g., 1 to 5 years)**	** *Health authorities/Governments (if not already instigated)* ** NAPs: Governments and health authorities across Africa must be committed to reducing the inappropriate use of antibiotics in hospitals. This will necessarily involve resources (technical/personnel and financial) to address current challenges, including currently limited activities surrounding ASPs, alongside building the necessary infrastructure, including electronic records, to routinely collect prescribing data. Electronic systems are essential to be able to routinely monitor prescribing against agreed prescribing/quality indicators. Potential prescribing/quality indicators: Agree with all key stakeholder groups on indicators for use within different hospitals in a country, building on the key principles for indicator development (Figure 1). Existing prescribing/ quality indicators (Table 2) can be used as a starting point. However, need to ensure that any agreed indicators are prioritized to prevent overload. Record keeping: The content and nature of any agreed indicator will depend on the nature of current patient record keeping, e.g., electronic or paper based, and how often the data are collected/ prescribing monitored. Indicators: Any agreed indicators need to be part of ongoing ASPs within hospitals. Training, resources and personnel must be devoted to instigating ASPs to enhance their chance of success (Box 1). If there are concerns with current limited activities within hospitals and a lack of knowledge and expertise within hospitals to undertake these, this can also be part of NAPs as well as ongoing ASP activities surrounding the WHO AWaRE list [56,64,204]. Culture differences: Any ASP activities must recognize that there are differences in cultures between countries. Any ASPs instigated will need to be country and culture specific, as well as multidisciplinary, to enhance their long-term sustainability among African countries [205,206], building on successful programs already instigated across Africa (Table 3). Key targets: For NAPs/ hospital ASPs across Africa include reducing the extent of prolonged prophylaxis to prevent SSIs given the extent of their overuse (Table 1), as well as the general overuse of antimicrobials in patients admitted to hospitals with COVID-19 [81,85,87,88]. Other key targets: Encouraging greater use of CST to guide future prescribing in hospitals. This depends on available and timely facilities and no/limited co-payments from patients for sensitivity testing [76]. National AMR surveillance facilities are growing across Africa, and this will continue alongside addressing infrastructure challenges, to help achieve NAP goals [9,53,207]. Robust guidelines: Need to be developed/ updated/disseminated for the management of key infectious diseases within hospitals among African countries, recognizing that active dissemination and communication of guidelines, as well as trust in those developing guidelines and their content, combined with their ease of use, are key to enhancing subsequent adherence rates [185,208,209,210,211,212]. This should include encouraging greater prescribing of ‘Access’ antibiotics where indicated [56,58], aided by the development of specific Apps to monitor the progress of ASPs and their impact on subsequent prescribing [135]. Monitoring prescribing: Monitoring against current guidelines and NAPs, enhanced by auditing, academic detailing and use of electronic record systems [208,211]. In addition, groups such as the Commonwealth Pharmacists developing and testing specific applications to assist with prescribing and ASPs [81,135,213]. AMS teams have a key role here along with Drug and Therapeutic Committees in hospitals [214,215]. This may mean increasing resources and training to ensure functioning AMS teams and DTCs where there are limited activities to date (Box 1) [216]. Adequate training: Ensure physicians, hospital pharmacists, microbiologists and other key healthcare professionals regarding antibiotics, AMR and ASPs are trained and continue to train post qualification (CPD) [130,217,218,219,220]. Increasingly, this is likely to involve hybrid learning building on the experiences during the COVID-19 pandemic [221]. Supply chains: Address supply chain bottlenecks which affect the routine availability of first-line (‘Access’) antibiotics and/or over-supply of ‘Watch’ antibiotics against current approved local guidelines. This is particularly important in low and low-middle-income countries where there can be considerable supply and access issues, e.g., Uganda. Strengthen prevention and detection of counterfeit/sub-standard antibiotics: This can be achieved through regional collaborative initiatives for capacity-building of regulatory authorities to enhance Good Manufacturing Practice (GMP), quality assurance, pharmacovigilance, and law enforcement, e.g., ZaZiBoNA which is an initiative among the SADC countries [222]. This builds on the recent WHO Lomé initiative [223]. ** *Healthcare professionals in hospitals* ** Ascertain current beliefs/knowledge: Regarding antibiotics, AMR, ASPs and NAPs especially where there are concerns with the current situation within hospitals and gaps in the knowledge of key HCPs. Multidisciplinary work: Work with Governments, health authorities and other key national organisations to develop (where pertinent) national/local evidence-based guidelines for important infectious diseases in hospitals, which are regularly updated and easily accessible increasingly through simple, easy to use applications and other systems [135,185]. This builds on current Pan-African initiatives [50,51,224]. Communication: Encourage physicians and other HCPs through auditing and other approaches to regularly consult their national/local guidelines about optimal treatment for their patients. This includes encouraging CST to reduce the extent of untargeted prescribing. Evidence: Microbiology, infectious disease specialists and other groups within hospitals actively producing and updating antibiograms to improve empiric prescribing whilst awaiting the results from sensitivity testing. Guidelines: Become actively involved in developing/reviewing national/local guidelines and ASPs, including the development of pertinent prescribing/quality indicators as part of hospital and NAP activities. This can also include ensuring, and be part of, active IPC groups within hospitals as well as Drugs and Therapeutic Committees where antibiotic use and availability is discussed building on concerns among African countries [214,216]. Training: Ensure students and HCPs continue training to improve their knowledge of antibiotics, AMR and ASPs building on national and international initiatives [46,225]. This can include improving communication skills with patients to address any concerns [226]. In addition, hospital pharmacists: (a) Education: Where necessary, enrol in courses to become knowledgeable about antibiotics and prescribing to assist physicians with their prescribing decisions; and in certain situations also potentially become prescribers. (b) Inclusion in ASPs: Must become actively involved with instigating and progressing ASPs in the hospital, building on activities within African countries such as South Africa and beyond [135,227]. (c) Provide educational support to physicians and other HCPs: Address any concerns regarding a lack of understanding or activities pertaining to antibiotics, AMR and ASPs within hospitals across Africa [129,130]. Outline appropriate antimicrobials to prescribe—especially pertinent where the main educational input on antibiotic prescribing in hospitals is via pharmaceutical companies and their literature [228,229]. (d) Guideline and prescribing/quality indicator development: Become involved with these initiatives as well as undertake PPS studies in hospitals. Promote targets for key quality improvement areas including antimicrobial use to prevent SSIs (Table 1) as well as documentation for the rationale for antibiotic prescribing, start and stop dates as well as active de-escalation from IV to oral antibiotics (Table 2). The development of an application and other electronic monitoring approaches should help [61,172], coupled with regularly feeding back concerns with antimicrobial usage patterns within hospitals to all key stakeholder groups and working with them on potential ways forward. (e) Antibiotic shortages: Actively work with key groups in hospitals and wider to proactively address possible antibiotic shortages where these occur. This means ascertaining key areas to address within current supply chains including ensuring timely payment of suppliers, checking suppliers have the capacity to deliver requested supplies as well as agreeing in advance potential therapeutic interchange recommendations ready for when the need arises [196,230].
**Long Term (5 to 10 years)**	**Potential long-term strategies include**: **Health authorities/Governments:** ○ NAPs: Regularly monitor antimicrobial utilization patterns across sectors as part of agreed NAPs across Africa [70]. This includes instigating electronic record systems within hospitals to track prescribing. ○ Antibiotic utilization: Instigate where pertinent additional multiple strategies to improve antibiotic utilization in hospitals, including the provision of necessary resources required for implementing ASPs/IPC committees in hospitals, routine CST and the development of hospital specific antibiograms, instigation of clinical decision support systems, and regular updating of guidelines. ○ Prescribing/quality indicators: Developing additional indicators/refining current indicators where pertinent to remain current as well as avoiding overloading HCPs. ○ Increasing investment in research: new and existing antimicrobials, diagnostic tools, and vaccines are all needed across Africa. **Physicians and other healthcare professionals:** ○ Educational activities: Regularly review current educational activities in medical/pharmacy/nursing schools regarding students’ knowledge of antibiotics, ASPs and AMR and keep up to date. ○ Key stakeholder groups: Keep engaging with key stakeholder groups to instigate additional ASPs where pertinent including where there is still extended antibiotic prescribing to prevent SSIs, there are concerns with lack of de-escalation of antibiotics and a continued lack of documentation in patients’ notes as part of ongoing NAPs. ○ Prescribing/quality indicators: Work with all key stakeholders to continue to develop/refine/update these—especially if outdated and where there is perceived overload. ○ Empiric prescribing: Continue to develop, update and communicate hospital antibiograms to improve empiric prescribing whilst awaiting CST results. ○ Regularly monitor prescribing activities: Quality improvement programs in hospitals including increased accountability of prescribers with a requirement to justify their treatment approach; Building restrictions for certain antibiotics where necessary based on the WHO AWaRe list and agreed quality indicators [56,58,66,231]. ○ Communication: Keep working with key stakeholders to enhance adherence to agreed national/local guidelines to improve patient outcomes and reduce AMR. ○ Hospital Pharmacists—Improve antibiotic utilization: Continue to monitor antimicrobial utilization patterns against agreed prescribing/quality indicators as part of agreed NAPs. In addition, regularly review therapeutic interchange policies for possible antimicrobial shortages as part of DTC and AMS programs. ○ Clinical Microbiologists/Laboratory scientists: Conduct and provide timely CST, including updating local antibiogram data in line with susceptibility patterns.

AMR: Antimicrobial Resistance; AMS: Antimicrobial stewardship; ASPs: Antimicrobial Stewardship Programs; CST: Culture and Sensitivity Testing; DTC: Drug and Therapeutic Committees; HCP: Healthcare Professional; IPC: Infection, Prevention and Control; NAP: National Action Plans for AMR; PPS: Point Prevalence Surveys; SSIs: Surgical Site Infections.

## Data Availability

Additional data are available on reasonable request from the corresponding author. However, all informational sources and papers have been extensively referenced. References [255,256,257,258,259,260,261,262,263,264,265,266] are cited in Appendix A.

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
