# Peer review of "Ongoing Efforts to Improve Antimicrobial Utilization in Hospitals among African Countries and Implications for the Future"

_antibiotics, 2022, doi:10.3390/antibiotics11121824_

Round 1

Reviewer 1 Report

The manuscript is well written but certain issues pointed out below need to be addressed.

  1. Line 67: do you mean “… AMR rates…”?
  2. Lines 88-90: The statement is confusing. 1.27 million deaths globally and then 4.95 million death in Africa? It’s not adding up.
  3. Line 91: “The high rate of AMR in African countries…” The authors should look at this paper below, particularly figure 2.

Klein, E. Y., Tseng, K. K., Pant, S., & Laxminarayan, R. (2019). Tracking global trends in the effectiveness of antibiotic therapy using the Drug Resistance Index. BMJ global health, 4(2), e001315. https://doi.org/10.1136/bmjgh-2018-001315.

They should rethink and revise this claim in line with the above article. If they insist on their claim, them they should substantiate it with citations.

  1. Lines 91-94: What is the correlation between high level of AMR in Africa and preponderance of HIV/AIDS, knowing that the condition/disease is of viral etiology?
  2. Lines 147-149: “This misuse of antibiotics….” I disagree with the authors. Are there no indications for the use of antibiotics in management of secondary bacterial infections in patients having viral diseases? Revise or delete that sentence.
  3. Lines 153-155: The problem of COVID-19 vaccine hesitance is not outside the study scope as it enhances the disease spread and facilities antibiotic misuse and hence AMR in Africa. I recommend that the authors access this recent article on vaccine hesitancy in African and then use it to slightly explain the nexus between vaccine hesitance and development of AMR.

Persisting Vaccine Hesitancy in Africa: The Whys, Global Public Health Consequences and Ways-Out—COVID-19 Vaccination Acceptance Rates as Case-in-Point. Vaccines 2022, 10, 1934. https://doi.org/10.3390/vaccines10111934

  1. The introduction is long with 129 citations. The authors could consider reducing the introduction so the readership doesn’t loss interest.
  2. Lines 236-237: Could this “highest utilization in Nigeria” be attributed to her high population estimated at about 208 million in 2022?
  3. Table 1: This Table is too long. The authors should insert a column to be titled “African regions” and classify the various countries listed into their respective regions; so that readers who are interested in a particular country could just go to the appropriate region and search out data on their country of interest without searching through the long Table.
  4. Line 270-273: Observe chronology. Mention Table 3 before Table 4.  
  5. Table 2: Comments for Table 1 (item 9) above apply.
  6. Line 297: Replace “turn-over” with “turnout”
  7. Line 328: do you mean …”of high priority”?
  8. Line 334-338: There are similar regulation/laws in other African countries but the problem is poor implementation. So, implementation is key not just mere introduction of a policy/regulation/law.
  9. Discussion: There is need to consider the roles of misuse and overuse of antibiotics in animal agriculture and the environment in the emergency, spread and control of AMR in humans. The authors may want to look at these papers: 

ü  Antimicrobial drug usage pattern in poultry farms in Nigeria: implications for food safety, public health and poultry disease management. Vet Ital. 2021 May 11;57(1):5-12. doi: 10.12834/VetIt.2117.11956.1.

ü  Antibiotic Resistance: One Health One World Outlook. Front Cell Infect Microbiol. 2021 Nov 25;11:771510. doi: 10.3389/fcimb.2021.771510

  1. Materials and methods: How did the authors choose the health facilities included in this study?

Author Response

Comment: The manuscript is well written but certain issues pointed out below need to be addressed.

Author Comment: Thank you for your kind words – appreciated.

1) Line 67: do you mean “… AMR rates…”?

Author Comment: Thank you – now addressed

2) Lines 88-90: The statement is confusing. 1.27 million deaths globally and then 4.95 million death in Africa? It’s not adding up.

Author Comment: Thank you for this comment. The reason for the 2 different global figures is that one refers to actual deaths attributable to AMR and the other associated deaths due to bacterial AMR and not deaths in Africa These are in fact lower and upper limits for the estimated burden of death due to AMR (plus the additional uncertainty intervals due to the methodology which is described in full in the original paper. We hope this is clear

3) Line 91: “The high rate of AMR in African countries…” The authors should look at this paper below, particularly figure 2.

Klein, E. Y., Tseng, K. K., Pant, S., & Laxminarayan, R. (2019). Tracking global trends in the effectiveness of antibiotic therapy using the Drug Resistance Index. BMJ global health4(2), e001315. https://doi.org/10.1136/bmjgh-2018-001315.

They should rethink and revise this claim in line with the above article. If they insist on their claim, them they should substantiate it with citations.

Author Comment: Thank you – we have now included this paper along with its limitations (e.g. included only one African country)  along with additional publications/ comments adding substance to our comments here and at the start of the introduction. We hope this is now OK.

4) Lines 91-94: What is the correlation between high level of AMR in Africa and preponderance of HIV/AIDS, knowing that the condition/disease is of viral etiology?

Author Comment: Thank you – we have now added to our comments for clarification, etc., and hope this is now acceptable.

5) Lines 147-149: “This misuse of antibiotics….” I disagree with the authors. Are there no indications for the use of antibiotics in management of secondary bacterial infections in patients having viral diseases? Revise or delete that sentence.

Author Comment: Thank you – I agree regarding bacterial and fungal co-infections. However, as stated, this applies to only a few in-patients with worrisome concerns for future AMR rates if this trend continues. We have added to our comments with additional references, etc., and hope this is now OK.  

6) Lines 153-155: The problem of COVID-19 vaccine hesitance is not outside the study scope as it enhances the disease spread and facilities antibiotic misuse and hence AMR in Africa. I recommend that the authors access this recent article on vaccine hesitancy in African and then use it to slightly explain the nexus between vaccine hesitance and development of AMR.

Persisting Vaccine Hesitancy in Africa: The Whys, Global Public Health Consequences and Ways-Out—COVID-19 Vaccination Acceptance Rates as Case-in-Point. Vaccines 202210, 1934. https://doi.org/10.3390/vaccines10111934

Author Comment: Thank you for this reference, which we have now included. We have also made further points regarding the need to address current hesitancy across Africa and the rationale. We have also included at the end of the Introduction why we did not look specifically at this point – especially since this has been addressed in other publications (this and others cited). We hope this is now acceptable.  

7) The introduction is long with 129 citations. The authors could consider reducing the introduction so the readership doesn’t loss interest.

Author Comment: Thank you for this. As seen, we have moved some sections to Supplementary Material which has reduced the number of initial references. However, we have needed to add in some additional references to address concerns from yourself and others. We hope this is now OK.

8) Lines 236-237: Could this “highest utilization in Nigeria” be attributed to her high population estimated at about 208 million in 2022?

Author Comment: Thank you for this. This actually refers to in-patient utilisation and does not correspond with the size of the country. We have made this point more strongly, and hope this is now OK.

9) Table 1: This Table is too long. The authors should insert a column to be titled “African regions” and classify the various countries listed into their respective regions; so that readers who are interested in a particular country could just go to the appropriate region and search out data on their country of interest without searching through the long Table.

Author Comment: Thank you for this. We have now moved Table 1 to Supplementary Material to reduce the length of the paper. If readers are interested in specific countries – they can just refer to these. We would like to keep the World Bank classification here for consistency to a previous SR that some of the co-authors were involved with. In addition, there have been concerns that it could be problematic for low- and low-middle income countries to undertake PPS studies due to financial and personnel issues – however we can now dispute this – with the published studies providing guidance to others! We hope this is now OK,

10) Line 270-273: Observe chronology. Mention Table 3 before Table 4.  

Author Comment: Thank you – now addressed.

11) Table 2: Comments for Table 1 (item 9) above apply.

Author Comment: Thank you – we would like to keep this classification for the same reasons as above. We hope this is OK again.

12) Line 297: Replace “turn-over” with “turnout”

Author Comment: Thank you – now amended.

13) Line 328: do you mean …”of high priority”?

Author Comment: Thank you – now amended

14) Line 334-338: There are similar regulation/laws in other African countries but the problem is poor implementation. So, implementation is key not just mere introduction of a policy/regulation/law.

Author Comment: Thank you – now addressed.  

15) Discussion: There is need to consider the roles of misuse and overuse of antibiotics in animal agriculture and the environment in the emergency, spread and control of AMR in humans. The authors may want to look at these papers: 

  1. a) Antimicrobial drug usage pattern in poultry farms in Nigeria: implications for food safety, public health and poultry disease management. Vet Ital. 2021 May 11;57(1):5-12. doi: 10.12834/VetIt.2117.11956.1.
  2. b) Antibiotic Resistance: One Health One World Outlook. Front Cell Infect Microbiol. 2021 Nov 25;11:771510. doi: 10.3389/fcimb.2021.771510

Author Comment: Thank you – As seen, we have included these references, and others, in the Introduction when talking about a One Health approach. However – we have not taken this further as this paper primarily deals with hospital antibiotic use. We hope this is OK.

16) Materials and methods: How did the authors choose the health facilities included in this study?

Author Comment: We did not choose health facilities or countries per se – this was entirely based on available publications. We have now made this point more strongly in the revised and hope this is now acceptable.

Reviewer 2 Report

The manuscript entitled “Ongoing Efforts to Improve Antimicrobial Utilisation in Hospitals among African Countries and Implications for the Futureinvestigated the antimicrobial utilization patterns via point prevalence surveys and antimicrobial stewardship programs among African hospitals.

The data collected from this study is important for scientists in the field to understand the picture of current antimicrobial utilisation patterns, surgical site infections’ antibiotic prophylaxis, and prescribing indicators in hospitals across Africa. Also, this study suggested several strategies to improve antimicrobial utilisation across Africa. The topic of this manuscript is appropriate for this type of journal. The study was well conducted and the manuscript was well written. However, some minor points are required to be improved as below:

 The study investigated the data collected from 12 countries in Africa. However, there are 46 countries across Africa. How about the data from the remaining countries? It would be important to clarify the data were collected from 12 countries across Africa in the main text. I suggest the authors provide a more accurate description “across 12 African countries” instead of “across Africa”.  There are five main geographical regions or subregions in Africa. It would be easier for the readers if the authors can include a figure of a map demonstrating the African countries and the number of hospitals involved in the study.

From line 151 to line 153: The authors mentioned the importance of vaccine equity. In recent years, several new antibiotics have been approved for treatment in developed countries such as ceftazidime/avibactam, cefiderocol, etc, but have not yet been applied in the remaining countries. Could the authors give more opinions about the equity of novel FDA-approved antibiotics in the discussion part?

Line 424-428: Please list the prescribing indicators in a supplementary table

Author Response

Comments and Suggestions for Authors

The manuscript entitled “Ongoing Efforts to Improve Antimicrobial Utilisation in Hospitals among African Countries and Implications for the Future” investigated the antimicrobial utilization patterns via point prevalence surveys and antimicrobial stewardship programs among African hospitals.

The data collected from this study is important for scientists in the field to understand the picture of current antimicrobial utilisation patterns, surgical site infections’ antibiotic prophylaxis, and prescribing indicators in hospitals across Africa. Also, this study suggested several strategies to improve antimicrobial utilisation across Africa. The topic of this manuscript is appropriate for this type of journal. The study was well conducted and the manuscript was well written. However, some minor points are required to be improved as below:

Author comments: Thank you for your kind comments regarding our paper – very much appreciated!

1) The study investigated the data collected from 12 countries in Africa. However, there are 46 countries across Africa. How about the data from the remaining countries? It would be important to clarify the data were collected from 12 countries across Africa in the main text. I suggest the authors provide a more accurate description “across 12 African countries” instead of “across Africa”.  

Author comments: Thank you for this. We did not in fact limit our search to these African countries – we were looking across Africa. We have now made this point more strongly in the revised paper to reduce any confusion since these were the African countries where we did in fact find publications regarding PPS studies (we were surprised just how many were published), studies looking at antibiotic use for SSIs (again surprised just how many were published), those dealing with QIs as well as ASPs (again surprised how many were published). Consequently, we would like to keep the wording as stated given the surprising amount of data that was available and sourced. We hope this is acceptable.

2) There are five main geographical regions or subregions in Africa. It would be easier for the readers if the authors can include a figure of a map demonstrating the African countries and the number of hospitals involved in the study.

Author comment: Thank you for this. However – the main aim of the paper was to provide input into the extent of PPS studies, studies looking at antibiotic use for SSIs, those dealing with Qis as well as ASPs to provide future guidance to all key stakeholder groups across Africa as there has been doubt regarding the extent of such activities – especially PPS studies and ASPs given resource concerns (e.g. the paper authored Cox et al that we mention). In addition, we broke the African countries down by World Bank Classification as there have been concerns that especially low- and low-middle income countries could not undertake PPS and ASPs due to resource concerns (echoed by Cox et al) – we have shown this not to be the case! Consequently, we would like to keep this classification (as per the previous systematic review undertaken by some of the co-authors) and not provide a map as such as this would be difficult to retain our classification and not add much to the paper/ may cause confusion. We have though (as seen) included the number of hospitals as described in the primary papers - in line with our previous systematic review of PPS studies - for those interested. However, we have not included the total number of hospitals included by each country given the duplication of hospitals in some studies. In addition – this was not the main emphasis of the paper – which was mainly to demonstrate that there have been a plethora of PPS studies conducted to date across Africa from low- to high upper-middle income countries providing exemplars to others.

Finally, we have moved Table 1 to Supplementary Material following concerns with one of the Reviewers about the length of the initial manuscript. We hope this is acceptable.

3) From line 151 to line 153: The authors mentioned the importance of vaccine equity. In recent years, several new antibiotics have been approved for treatment in developed countries such as ceftazidime/avibactam, cefiderocol, etc, but have not yet been applied in the remaining countries. Could the authors give more opinions about the equity of novel FDA-approved antibiotics in the discussion part?

Author comments: Thank you – we have now added comments on this in the revised paper – linking the 2 together. We hope this is now acceptable.

4) Line 424-428: Please list the prescribing indicators in a supplementary table

Author comments: Thank you for this. We have moved some parts of the Introduction in this area to the Appendix to help focus the Introduction (in line with comments from another Reviewer). We would though like, if you are agreeable, to keep the Prescribing Indicator Table in the main part of the paper as it shows that there are extensive activities across Africa to improve future antimicrobial prescribing in hospitals (which will no doubt be a surprise to a number of key stakeholders if this manuscript is accepted for publication) which can feed into future ASPs. We hope this is acceptable to you.